# SALL4 and microRNA: The Role of Let-7

**DOI:** 10.3390/genes12091301

**Published:** 2021-08-24

**Authors:** Jun Liu, Madeline A. Sauer, Shaza G. Hussein, Junyu Yang, Daniel G. Tenen, Li Chai

**Affiliations:** 1Department of Pathology, Brigham & Women’s Hospital, Boston, MA 02115, USA; jliu71@bwh.harvard.edu (J.L.); masn8c@health.missouri.edu (M.A.S.); y_junyu@yahoo.com (J.Y.); 2Harvard College, Cambridge, MA 02138, USA; sghussein@college.harvard.edu; 3Cancer Science Institute, National University of Singapore, Singapore 117599, Singapore; 4Harvard Stem Cell Institute, Harvard Medical School, Boston, MA 02115, USA

**Keywords:** SALL4, Let-7, LIN28, microRNA, miR-98, non-small cell lung cancer (NSCLC), hepatocellular carcinoma (HCC)

## Abstract

SALL4 is a zinc finger transcription factor that belongs to the spalt-like (SALL) gene family. It plays important roles in the maintenance of self-renewal and pluripotency of embryonic stem cells, and its expression is repressed in most adult organs. SALL4 re-expression has been observed in different types of human cancers, and dysregulation of SALL4 contributes to the pathogenesis, metastasis, and even drug resistance of multiple cancer types. Surprisingly, little is known regarding how SALL4 expression is controlled, but recently microRNAs (miRNAs) have emerged as important regulators of SALL4. Due to the ability of regulating targets differentially in specific tissues, and recent advances in systemic and organ specific miRNA delivery mechanisms, miRNAs have emerged as promising therapeutic targets for cancer treatment. In this review, we summarize current knowledge of the interaction between SALL4 and miRNAs in mammalian development and cancer, paying particular attention to the emerging roles of the Let-7/Lin28 axis. In addition, we discuss the therapeutic prospects of targeting SALL4 using miRNA-based strategies, with a focus on the Let-7/LIN28 axis.

## 1. Introduction

SALL4 is a zinc finger transcription factor that belongs to the spalt-like (SALL) gene family [1]. In humans, the SALL4 gene is located at 20q13.2, with three known alternative splicing isoforms named SALL4A, SALL4B, and SALL4C [2,3]. SALL4 plays a significant role in the pluripotency of embryonic stem cells through maintenance and self-renewal [4]. Several studies have found that SALL4 works with key pluripotency transcription factors OCT4, SOX2, and NANOG, underlying the mechanism of SALL4 function in embryonic stem cells (ESCs). 

In the adult organism, SALL4 expression is silenced in most organs [5,6,7] and is largely restricted to germ cells, with only one exception of human CD34+ hematopoietic stem cells (HSPCs) [8]. SALL4 re-expression has been observed in different types of human cancers and is regarded as an important biomarker [9]. Furthermore, induction of SALL4 expression in murine models can induce myelodysplastic syndrome (MDS) and progression to Acute Myeloid Leukemia (AML), demonstrating that SALL4 re-expression had oncogenic function [10]. Dysregulation of SALL4 can be observed in multiple human cancer types, including germ cell tumors, leukemia, hepatocellular carcinoma (HCC), colorectal carcinoma (CRC), breast cancer, and lung cancer [11,12,13,14,15]. Up-regulation of SALL4 promotes proliferation, metastasis, and drug resistance of cancer cells [16,17,18]. The tumor suppressor gene PTEN has been identified as a key downstream target of SALL4, as SALL4 directly interacts with the nucleosome remodeling and histone deacetylation (NuRD) complex and can be recruited to the promoter region of PTEN and other genes, inducing transcriptional repression [19]. Besides its repression function, SALL4 can also activate gene expression. In AML, SALL4 interacts with the MLL complex and induces the up-regulation of HOXA9, leading to tumorigenesis [20]. In addition, SALL4 can enhance the expression of the oncogene c-MYC by directly binding to its promoter region [21], as well as genes involved in oxidative phosphorylation [22]. While the biological functions and downstream targets of SALL4 have been well characterized, the upstream regulators of SALL4 remain understudied.

MicroRNAs (miRNAs) are a group of small non-coding RNAs that act as post-transcriptional regulators. MiRNAs can induce mRNA degradation or translational repression of their targets by directly binding to the 3′untranslational region (3′UTR) [23]. miRNAs affect a wide array of cellular processes including self-renewal, cell proliferation, cell cycle, migration, and apoptosis [24]. MiroRNAs have become of interest for their ability to affect small scale cellular processes, and because of identified dysregulation seen in tumorigenesis [25]. Notably, miRNAs can act as both oncogenes and tumor suppressors. For example, the liver specific miRNA miR-122 is downregulated in HCC and could repress tumorigenesis by targeting the TGF-beta pathway [26], and the Let-7 family members are frequently identified as tumor suppressors by inhibiting oncogene c-Myc [27]. On the other hand, miR-221 is reported to be up-regulated in lung cancer and to enhance tumor development by targeting the tumor suppressor gene PTEN [28]. Due to their ability to regulate multiple targets in a tissue specific manner [29,30], and also recent advances in systemic and organ specific miRNA delivery mechanisms, miRNAs have emerged as promising therapeutic targets for cancer treatment.

In this systematic review, we summarize the current knowledge of SALL4 and miRNAs functions in development and cancer. We focus on members of miRNAs that target SALL4, with particular attention to the emerging roles of the Let-7/Lin28 axis in SALL4 regulation in the context of normal development and cancer. In addition to mechanistic studies in cell culture and in vivo models, we also examine clinical research studies showing diagnostic or prognostic correlates of microRNAs in tumor types in which SALL4 reactivation is known to play a pathogenic role. In addition, we review the evidence of SALL4 as a miRNA regulator and discuss therapeutic prospects of targeting SALL4 using miRNA-based strategies, in particular targeting the Let-7/LIN28 axis. 

## 2. Materials and Methods

### 2.1. Inclusion and Exclusion Criteria

Studies describing the interactions between microRNA and SALL4 in cancer or development were included. Eligibility criteria were studies that included characterization of a microRNA, SALL4, and their interactions. We included original research spanning a wide spectrum, including functional studies using cell lines or animal models, clinical research studies using patient samples, epidemiological studies, and in silico predictive studies of microRNA targets. Clinical studies include prospective and retrospective cohort studies, case series, meta-analysis and clinical trials. Case reports were excluded. We included studies published up to July 2021.

### 2.2. Search Strategies

We conducted literature searches with PubMed up to July 2021. We used medical subject headings (MeSH) terms “microRNAs” or “SALL4” without using language restrictions. Two independent reviewers assessed titles and abstracts of the papers that are initially identified by the search. We obtained potentially relevant papers and conducted subsequent full text screening. We also conducted a manual search based on our knowledge of a relevant article on BioRxiv which was accepted for publication in a peer-reviewed journal indexed by PubMed, as well as manual search of the literature based on the cited articles of the relevant full text articles. Any disagreement between investigators was resolved by consensus. A flow chart documenting the article screening process is listed in Figure 1. 

## 3. Results

### 3.1. Interaction between LIN28/Let-7 and SALL4 Regulates Normal Mammalian Development 

The Let-7 family is a diverse group of microRNAs that includes 13 members which encode 9 unique mature microRNAs [1]. The members of the Let-7 family of miRNAs differ by 1-4 nucleotides and are highly conserved across species in sequence and function. The seed region of 3′GAUGGAGU5′ targets several oncogenes by directly binding to their 3′ UTR regions [1]. Although Let-7a1, a2, a3 are encoded by different transcripts, they produce identical mature sequence; let-7f1 and f2 transcripts also give rise to identical mature miRNA [1]. Mir-98 is considered a member of the let-7 family based on its seed sequence. LIN28 encodes an RNA-binding protein. Activation of either of the two isoforms, LIN28A or LIN28B, can lead to post-transcriptional downregulation of the Let-7 microRNA family. Specifically, LIN28A prevents microRNA biogenesis by directly binding to the terminal loop of Let-7’s precursor molecule. This binding leads to Terminal Uridylyl Transferase (TUTase) ZZCHC11 which polyuridylates pre-Let-7 [2]. LIN28 recognition by Let-7 microRNA is mediated by targeting of both the zinc knuckle domain (ZKD), which recognizes a GGAG-like element in the microRNA precursor, as well as the cold shock domain (CSD). The mechanisms of CSD domain binding on the microRNA have recently been characterized, and the (U)GAU motif was shown to be the target sequence [3]. This motif separates the let-7 microRNAs into two subclasses, including precursors with both CSD and ZKD binding sites (CSD+) and precursors with ZKD but no CSD binding sites (CSD-). In vivo recognition of CSD+ precursors by LIN28 have been shown to be more efficient, which corresponds to stronger suppression of these microRNAs in LIN28-activated cells and cancers. CSD+ Let-7 precursors include pre-let-7b, pre-let-7d, pre-letf-1, pre-let-7g, and mir-98. The binding between LIN28 and mir-98 CSD domain has been validated in vivo: loss-of-function mutant of pri-let-7 hairpins from miR-98 where the (U)GAU motif was mutated shows greatly reduced the amount of LIN28 protein binding [3].

Recent studies demonstrate that the LIN28/Let-7 axis has important functions in both normal organismal development and tumorigenesis, at least in part acting through SALL4. In development, the Let-7 family of miRNAs is widely observed among differentiated tissues and is important during ESC differentiation [4,5,6]. Overexpression of Let-7 could restore differentiation potential by repressing the self-renewal process in DGCR8-knockout mouse ESCs [7]. Let-7 promotes differentiation of ESCs by directly targeting and repressing the expression of several stemness factors, including SALL4 and c-Myc, both of which have proven Let-7 binding sites in their 3′UTR region. This finding is consistent with the stemness maintenance function of SALL4 in ESCs. 

During embryonic development, the Lin28/Let-7 axes serve as highly conserved, important heterochronic genes which regulate the timing of morphogenesis and govern body plan formation [8,9]. Through translational inhibition of Lin28a/b by the Let-7 microRNA (miRNA) family leads to cellular differentiation. A recent study found that overexpression of Lin28a increased caudal vertebrae number and tail bud cell proliferation, and knockout of Lin28 decreased this proliferation. Manipulating Lin28a and Let-7 led to opposite effects, suggesting that Lin28/Let-7 may regulate tail length through heterochrony of the body plan [10]. The authors used TargetScan to highlight predicted Let-7 targets [11]. Interestingly, they found *Sall4*, an important regulator of key pluripotency genes in stem cells, was modestly upregulated under Lin28a overexpression (and putative downregulation of Let-7). The upregulation of SALL4 was validated via immunofluorescence. Based on these data, the authors proposed that a number of Let-7 targets, including SALL4, may contribute to this phenotype during embryonic development by influencing canonical signaling [10].

### 3.2. The Let-7 microRNA Contributes to Tumorigenesis 

Tumorigenesis frequently hijacks developmental mechanisms, so it is perhaps not surprising that the Lin28/Let-7 axis also plays important roles in cancer. Functionally, the activation of either LIN28A or LIN28B and the subsequent decrease in Let-7 microRNAs can lead to overexpression of Let-7’s oncogenic targets, such as MYC, RAS, HMGA2, and BLIMP1 [1]. Furthermore, mouse models show an accelerated tumorigenesis in a Let-7 dependent manner with ectopic expression of LIN28. Based on these observations, Lin28A/B have been proposed as “proto-oncogenes”, whereas Let-7 microRNAs as “tumor suppressors”. Studies have shown abnormal Let-7 expression levels in various human cancer tissues. The extent to which Let-7 dysregulation contributes to tumorigenesis and progression in each of the tumor types, as well as the prognostic or therapeutic implications are still active areas of research investigation. For example, studies show that Let-7a overexpression induces apoptosis and cell cycle arrest of HCC cells; and that Let-7a inhibits self-renewal of HCC stem-like cells through regulating the EMT and the Wnt signaling pathway [12]. Liang et al. demonstrated that Let-7e downregulation inhibits proliferation and may affect apoptosis and autophagy of HCC cells [13].

In a preliminary in silico analysis conducted via TargetScan, we found that the seed sequences of all the Let-7 members matched to the same SALL4 mRNA 3′UTR binding site at positions at 1335-1342. Parallel to the well characterized interaction between the Let-7 microRNA family and SALL4 during development, several recent studies demonstrate that Let-7/miR-98 may act as a tumor suppressor via targeting SALL4 in HCC, non-small cell lung cancer (NSCLC), and ovarian carcinoma [14,15,16]. In addition, there is an emerging role for Let-7 and SALL4 in glioma [17,18,19,20], and we hypothesize that Let-7 and SALL4 interaction may be biologically relevant in this context as well. 

### 3.3. Let-7 microRNA and SALL4 Interaction in HCC

HCC has a relatively poor prognosis due to a combination of factors, including difficulty in detecting HCC early [21], as well as poor treatment options. This is especially challenging in some developing countries, in which the exposure risks to hepatitis viruses are high and vaccination rates relatively low. Overexpression of microRNAs has been found to correlate with burden of disease in HCC and remains an active area of research. Several groups investigated Let-7 and its potential implications in HCC: some Let-7 members appear to function as oncogenes whereas other members act as tumor suppressors. Jiang et al. showed that Let-7a and Let-7d levels were significantly correlated with venous invasion in HCC models [22]. Another group confirmed Let-7a levels correlate with venous invasion, and further identified high Let-7c correlating with a shorter overall survival postoperatively [23]. Liang et al. showed that downregulation of Let-7e inhibited cell density both through inhibition of cell proliferation and colony forming ability in HCC cells [13].

In contrast, the Let-7 member miR-98 has gained attention for its potential anti-tumorigenic effects in a wide array of cancers including HCC. Generally speaking, increased miR-98 levels downregulate many cellular processes necessary for HCC cell survival, migration, and invasion [15,24,25]. A recent study found that mir-98 levels are markedly downregulated in HCC and its overexpression inhibits proliferation, migration, invasion, and EMT of HCC cells [26]. SALL4 was further identified as a direct target of miR-98. SALL4 is frequently re-expressed or “reactivated” in HCC and associated with a poorer prognosis [27,28,29]. The expression levels of SALL4 and miR-98 were inversely correlated in HCC tissues, suggesting that upregulation of SALL4 may follow downregulation of miR-98 in HCC [26]. Furthermore, using a mouse xenograft model, miR-98 inhibited the growth of HCC in vivo. This was demonstrated by significantly lower tumor volume and weight in the miR-98 treated group compared to control. The authors also showed that overexpression of miR-98 significantly inhibited SALL4 expression in mRNA and protein levels in HCC cell lines [26]. 

In addition to SALL4, other mechanisms mediating the tumor-suppressive effect of miR-98 in HCC have been proposed. Li et al. found that miR-98 can act on tumor associated macrophages (TAMs) to alter polarization, thereby alerting cytokine expression intratumorally [24]. MiR-98 and its mimics could suppress cell migration signals and epithelial mesenchymal transition from TAMs [24]. Additionally, miR-98 targets the IL-10 gene and can lead to increased IL-10 levels and decreased inflammatory cytokines inducing cell migration and invasion in HepG2 cell lines [24]. A further study showed that miR-98 can also upregulate P53 expression and downregulate Bcl-2, ang-1, and FGF-1 gene expression, providing insight into its ability to induce cell apoptosis and decrease EMT signals [30].

### 3.4. Let-7 microRNA Regulates SALL4 in NSCLC

Lung cancer is one of most prevalent malignancies worldwide, and NSCLC accounts for approximately 85% of lung cancers. The overall survival of patients with late-stage NSCLC remains unsatisfactory despite all the available medical and surgical management. A recent meta-analysis demonstrated that low Let-7 expression is a significant predictor of worse prognosis in patients with lung cancer [31]. Many mechanisms have been proposed for a protective role of Let-7 in lung cancer. The Let-7 member miR-98 inhibited the malignant phenotypes of NSCLC cells through directly targeting ITGB3 and PAK [16]. Downregulation of Let-7 correlates with accumulation of insertions of the transposable element L-1 in human lung cancer, likely causing genome instability and driving tumor genome evolution [32]. Let-7 can bind directly to L1-mRNA. Studies on cultured tumor cells showed that this interaction impairs L1-ORF2p translation and reduces L1 retrotransposition. Altogether, these results identified Let-7’s ability to maintain genome integrity. Furthermore, the mechanism behind downregulation of Let-7 miRNAs can lead to unbridled L1 activity in tumors. 

Recent studies suggest that SALL4 is important for lung cancer pathobiology and may serve as a potential therapeutic target for the diagnosis and treatment of lung cancer. EGFR mutated lung tumors expressed SALL4 to a significantly higher degree than non-tumor tissue. Furthermore, lung cancer patients with increased SALL4 expression had poorer prognosis post-surgical resection when compared with low SALL4 expressing patients. This may be in part due to decreased migration, invasion, and metastasis of lung cancer cells and increased sensitivity to Erlotinib in EGFR mutated cells seen with knockdown of SALL4 [33]. Another study compared SALL4 levels between patients prior to adjuvant chemotherapy and found higher SALL4 expression in patients with recurrent cancer when compared with those who were disease-free. They also showed the time until disease recurrence was shorter in the high SALL4 expressing patients [34].

Against this backdrop, it is not surprising that a recent study uncovered direct regulation of SALL4 by miR-98 in NSCLC. Consistent with prior research, Liu et al. showed that SALL4 was significantly upregulated and miR-98 was significantly downregulated in NSCLC tissues compared to normal lung tissues [16]. Furthermore, poorer differentiation and more advanced clinical stages were seen in the tissues expressing lower levels of miR-98. Using NSCLC cell lines, the authors showed that restoring miR-98 expression was able to significantly decrease the proliferation, migration, and invasion of NSCLC A549 and H1229 cells. Importantly, the authors identified SALL4 as a key target gene of miR-98, showing that expression of SALL4 protein was negatively regulated by miR-98. Study of A549 and H1229 cell lines showed heightened cell proliferation, migration, and invasion when SALL4 was overexpressed, identifying SALL4s ability to reverse the effects of miR-98. These results suggest that miR-98 acts as a tumor suppressor in NSCLC cells by inhibiting the protein expression SALL4 and highlights the importance of a Let-7/SALL4 axis in NSCLC [16]. In addition to liver cancer and NSCLC, both SALL4 and Let-7 family microRNAs have been shown to play important roles in the pathogenesis of multiple cancer types, such as glioma, ovarian cancer, colorectal cancer, breast cancer, and myeloid neoplasms [1,35]. However, further studies are necessary to explore the direct regulation of SALL4 by Let-7 family members in these cancer types.

### 3.5. Interaction between Other microRNAs and SALL4 in Cancer Biology

Apart from Let-7 microRNAs, other microRNAs have been shown to regulate SALL4 or are regulated by SALL4 in various types of tumors. Existing evidence of SALL4 interacting microRNAs is summarized in Figure 2, with experimental details of select microRNAs highlighted below. We found a surprisingly high number of published studies that explored the regulation of SALL4 by microRNAs in glioma. Gliomas are a group of primary brain tumors that are thought to derive from neuroglial stem or progenitor cells, which includes astrocytic, oligodendroglial, or ependymal tumors. The clinical course and prognosis vary tremendously among the different variants of gliomas. Glioblastoma multiforme (GBM) has an average 10-year survival rate of 2.6%, and it remains one of the deadliest cancers with an abysmal prognosis [36]. MiR-107 is a broadly explored miRNA, which can act as a tumor suppressor or oncogene, depending on different cancer types [37]. In glioma tissues and cell lines, miR-107 exhibits reduced expression level. Up-regulation of miR-107 significantly induced apoptosis and inhibited tumor growth both in culture and in vivo [17]. Further studies demonstrated that miR-107 could directly target and repress SALL4 expression. SALL4 overexpression could rescue apoptosis induced by miR-107, and its level was negatively correlated with miR-107 in glioma tissues. These results highlight the significant role of SALL4 in glioma development and identify miR-107 as a suppressor of SALL4. Many other microRNAs have been proposed to regulate glioma tumorigenesis via SALL4, such as miR-219, miR-16, and miR-103/195 [17,26,38,39], and a complete list is summarized in Figure 2. Noteworthily, most of the other microRNA/SALL4 interactions are only supported by individual publications and await further validation, except for miR-219 which has also been shown to regulate SALL4 in colon cancer, and miR-16 in gastric cancer in addition to glioma [40,41]. MiR-33b regulates SALL4 in both breast cancer and HCC [42,43]. 

miR-219-5p is another broadly studied miRNA in cancers [44,45,46]. Consistent with its demonstrated anti-tumor role in hepatocellular carcinoma, papillary thyroid carcinoma, and glioblastoma, miR-219-5p also exhibited tumor suppressor activity in colon cancer by inducing apoptosis and reducing drug resistance [40]. SALL4 was identified as the main target of miR-219-5p in colon cancer. During the development of colon cancer, miR-219-5p expression decreased, leading to a significant up-regulation of SALL4, which could repress cancer cell apoptosis and enhance drug resistance.

Human miR-33b is located in the intron of the SREBP-1 gene [47,48]. A recent study revealed the role of miR-33b in tumorigenesis of breast cancer [42]. miR-33b was down-regulated in breast cancer tissues compared to normal tissues and reversely correlated with tumor progression. Combining miRNA targets prediction database and qRT-PCR validation, the authors identified SALL4 as one of top miR-33b targets in breast cancer. Intriguingly, miR-33b could repress self-renewal of breast cancer cells, which is one of the main functions of SALL4. Meanwhile, through inhibition of SALL4, miR-33b also suppressed metastasis of breast cancer in vivo. These findings demonstrate that miR-33b can regulate both stemness and metastasis in breast cancer by targeting SALL4.

Very little is known about the downstream miRNA targets of SALL4. A recent study has discovered a link between SALL4 and PD-L1 in HBV-induced HCC through miR-200c regulation [49]. Chronic hepatic viral infection is one of the major contributors to the development of HCC. PD-1 is an inhibitory receptor for T cell activation, which is critical for clearance of viral infection [50,51]. PD-L1 is the ligand of PD-L and can be expressed on HCC cells. The interaction between PD-1 and PD-L1 can lead to T cell exhaustion. In this study, a reverse correlation between PD-L1, or SALL4 and miR-200c was observed with corresponding survival trends in HCC patients. The authors further propose that in an HBV setting, SALL4 is reactivated by the virus which leads to decreased miR-200 [49]. Since miR-200 downregulates PD-L1, the net results of this process lead to a higher PD-L1 expression, apoptosis of CD8+ T cells, which may contribute to the worse outcome of HCC. In addition to miR-200c, SALL4 has also been shown to regulate the expression of miR-146a in HCC [52]. Treatment with HCC-derived exosomes could remodel M2 macrophages, at least in part by activating the NFkB signaling pathway. miR-146a is highly enriched in the HCC exomes, and its expression is under the direct regulation by SALL4 via binding to the microRNA’s promoter region. Although the authors showed that the expression of miR-146a recapitulates the functional consequences of HCC exome treatment by promoting M2 macrophage remodeling and leading to changes in the expression of IFN-γ, TNF-α, PD-1, and CTLA-4, the precise mechanisms (e.g., the involvement of the NFkB pathway) require further elucidation. 

All microRNAs that are shown to regulate SALL4 and are regulated by SALL4 are summarized in Table 1, including levels of evidence based on experimental designs, seed sequence information, binding location on the 3′UTR (if applicable), and bibliography. Other less well characterized microRNAs include miR-15a-5p, miR-485-5p, miR-3622a-3p, miR-188-5p, miR-181b, miR-296-5p, miR-3619-5p, and miR-146a-5p [52,53,54,55,56,57,58,59]. 

### 3.6. Harnessing microRNA for SALL4 Targeting in Cancer Treatment

Previously, multiple strategies have been proposed to target SALL4 for cancer therapies. A new trend in cancer therapy is to target a transcriptional factor (TF) and/or its interaction with epigenetic complexes to affect the transcriptome unique to malignant cells. Along this direction, much effort has been applied to block the interaction of SALL4 with other proteins. The mechanism of SALL4 in tumorigenesis, in part, is through its interaction with Nucleosome Remodeling Deacetylase (NuRD) complex to repress its target genes such as PTEN [61]. Blocking the interaction between SALL4 and RBBP4, a component of NuRD complex, by a peptide can reduce its function in tumorigenesis [29,62,63]. Another approach is to target SALL4 function by HDAC inhibitor Entinostat [64] or inhibitors to oxidative phosphorylation pathways [65].

Harnessing endogenous SALL4-microRNA interaction may be a new therapeutic avenue. Recently, Yang et al. showed that Entinostat, which is well known to negatively affect SALL4 expression, acts through a microRNA, miRNA-205, both in culture and *in vivo* [60]. Decreased SALL4 expression leads to decreased cellular viability and increased apoptosis in vivo consistent with known effect of SALL4 on the expression of oncogenes and tumor suppressor genes, but the study did not further investigate the downstream effectors of SALL4 in this context (Figure 3). 

With the advancement of tissue specific microRNA delivery mechanisms and the expansion of literature on SALL4-targeting microRNAs in various cancer types, we propose that targeting SALL4 with microRNA-based therapeutics may be a new direction in overcoming SALL4 reactivation. RNA-based cancer treatments are currently an area of active clinical investigation. According to Clinicaltrial.gov, a website that collects data on privately and publicly funded clinical trials, there are currently 1040 clinical trials ongoing or recruiting for microRNA research, among which 324 are related to cancer treatment.

As described in the sections above, SALL4 plays a significant role in the pathophysiology of a significant number of cases of HCC [29]. Fortuitously, there has been an explosion of therapeutic development of liver-targeted RNA delivery mechanisms, such as antisense oligonucleotides (ASOs), short interfering RNAs (siRNAs), and microRNAs, using nanoparticles and exosomes expressing ligands to liver specific receptors [66]. Proof-of-principle preclinical and clinical studies have demonstrated promising results for the treatment of metabolic liver diseases such as hereditary hemochromatosis, alpha 1 antitrypsin deficiency, and nonalcoholic fatty liver disease (NAFLD) [67,68]. A promising area of research would be to explore the potential of using mimics of Let-7 microRNAs, such as miR-98, and other SALL4 targeting microRNAs for treatment of HCC in pre-clinical and clinical models. With regards to Let-7 microRNAs specifically, one may also take advantage of the antagonistic relationship between LIN28 and Let-7 for drug development. LIN28 can suppress Let-7 microRNAs maturation thereby promoting transformation in malignancy. Previously, Wang et al. sought to identify small molecules that could restore mature Let-7 levels by inhibiting LIN28 using a high-throughput screening strategy [69]. In addition, knockdown of LIN28 restores Let-7 levels which led to the suppression of Wilms tumor and hepatoblastoma in mouse models [70,71]. One added benefit of LIN28 based therapeutics is the putative lack of toxicity: while developmental suppression of LIN28 expression during the transition from fetal to adult life is critical, very low levels of LIN28 found in normal adult tissues should limit its toxicity [69]. Figure 3 summarizes proposed microRNA based SALL4 targeting pathways and downstream regulators. When Let-7/miR-98 is upregulated with LIN28 inhibitors, decreased SALL4 expression causes decreased expression of MMP2/9, Fibronectin, N-cadherin, and increased expression of E-cadherin, which correlate with decreased migration/invasion and EMT in an HCC in vivo cancer model. Increased miR-98 and repression of SALL4 expression also leads to decreased tumor cell proliferation in vivo and this is presumably due to alteration of downstream expression of SALL4-regulated oncogenes and decreased tumor suppressor gene expression (Figure 3). 

Despite recent advances in molecular profiling and therapeutic development, treatment options for glioblastoma are still limited. Recently, Dr. Peruzzi’s group from Brigham and Women’s hospital, Boston, successfully developed a viral vector that enables the delivery of therapeutic microRNA to GBM cells in a murine glioblastoma model, leading to five-fold increase in survival when combined with chemotherapy [72,73]. Given that multiple microRNAs are found to regulate SALL4 in glioma, further studies should explore the use of microRNA-based therapy for SALL4 inactivation in the context of glioma. Although no study explored the direct interaction between SALL4 and Let-7 family members in glioma, recent studies show consistent downregulation of Let-7 microRNA in glioma. For example, the expression level of miR-98 [19,74], Let-7b [20], and Let-7g [19] in human glioma (or GBM) are all significantly lower than that in normal brain tissues, and lower Let-7 in GBM is associated with poorer prognosis in GBM [20]. One recent study provided support for therapeutic targeting of the LIN28/Let-7 axis in GBM: silencing advillin (AVIL), a gene that is overexpressed in GBM and correlated with worse prognosis. The silencing of AVIL leads to near eradication of glioblastoma cells in culture, and in vivo xenograft mice modes dramatically inhibit glioblastoma cells, while having no effect on the normal control cells [75]. The tumorigenic effect of AVIL is partly mediated by FOXM1. FOXM1 is a known regulator of LIN28B, whose main function is inhibiting Let-7 family microRNAs [75]. 

## 4. Conclusions

The oncofetal protein SALL4 plays significant roles in embryonic development, stem cell maintenance, and tumorigenesis. Upstream mediators of SALL4 have been explored to some extent, including activation by the WNT pathway through the TCF/LEF consensus sequence in the SALL4 promoter region, and repression by epigenetic DNA methylation in the SALL4 genome. More recently, miRNAs that target SALL4 have emerged as important regulators. In this systematic review, we summarize the current knowledge of SALL4 and miRNA functions in development and cancer, with a focus on the Let-7/Lin28 axis. 

Multiple miRNAs, including the Let-7 member miR-98, have been reported to target SALL4 and regulate tumor cell growth, apoptosis, and metastasis in various tumor types. As all SALL4 isoforms share the same 3′UTR sequence, miRNAs have their unique property to target the two most studied isoforms, SALL4A and SALL4B, at the same time, as SALL4B is an internal spliced variant of SALL4A. Meanwhile, with the expansion of our knowledge about miRNA regulation machinery and advances in RNA therapeutic delivery, a better understanding of the microRNA/SALL4 interaction in tumor biology may help guide the development of targeted treatment. As an important transcription factor, SALL4 may also regulate the expression of certain miRNAs through transcriptional or post-transcriptional mechanisms, and further studies are needed to explore this understudied area. 

## Figures and Tables

**Figure 1 genes-12-01301-f001:**
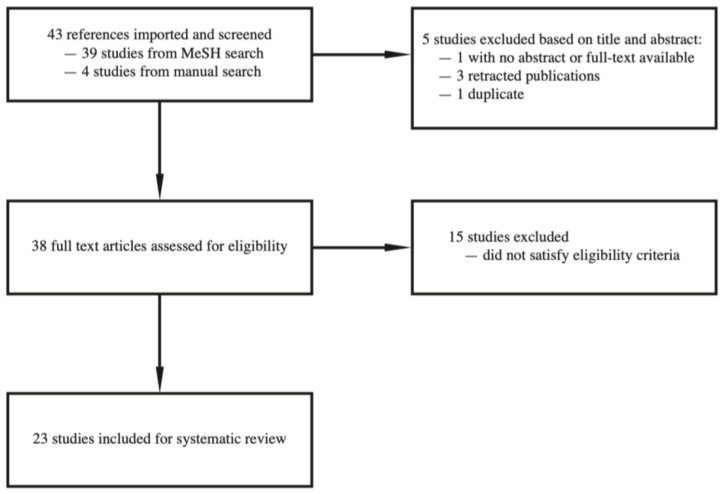
Flow chart of article screening, inclusion, and exclusion.

**Figure 2 genes-12-01301-f002:**
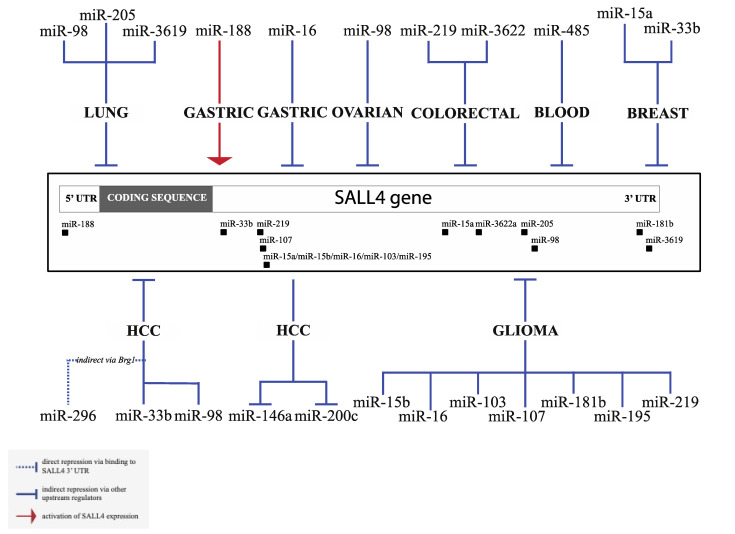
Summary of Spalt-like (SALL4)-interacting microRNAs: Multiple microRNAs have been shown to regulate SALL4 in a variety of cancer types, including glioma, acute myeloid leukemia (AML), hepatocellular carcinoma (HCC), gastric cancer, ovarian cancer, lung cancer, colorectal cancer, and breast cancer. In addition, SALL4 acts as an upstream regulator of the expression of miR-200c and miR-146a in HCC.

**Figure 3 genes-12-01301-f003:**
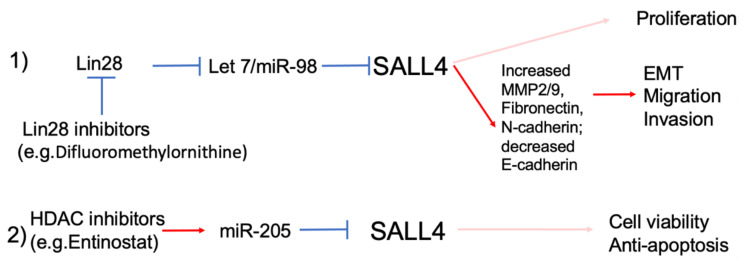
Strategies in targeting miR/SALL4 axis in cancer. We can inhibit SALL4 protein expression through miRs by (1) using LIN28 inhibitors to block its repression on Let 7/miR-98 which in turn decreases SALL4 expression (2) use of epigenetic drugs such as HDAC inhibitor Entinostat to upregulate miR-205, which in turn suppresses SALL4 expression. Solid lines represent experimentally validated epistatic relationships.

**Table 1 genes-12-01301-t001:** Summary of all microRNAs known to regulate SALL4 and are regulated by SALL4.

	Cancer	microRNA	Method(s)	Seed Sequences	Position of SALL4 3′UTR	Reference
microRNAs targeting SALL4	Breast	miR-15a	In Silico, In Vitro, In Vivo	AGCAGCA, SALL4 3′UTR	175−181, 967−973	[59]
		miR-33b	In Silico, In Vitro, In Vivo	UGCAUU, SALL4 3′UTR	23−29	[42]
	Leukemia	miR-485	In Silico, In Vitro	No information		[58]
	Colorectal	miR-219	In Silico, In Vitro	UGAUUGU, SALL4 3′UTR	156−161	[40]
		miR-3622a	In Silico, In Vitro, In Vivo	CACCUGA---UCCC, SALL4 3′UTR	1079−1092	[57]
	Gastric	miR-16	In Silico, In Vitro	AGCAGCA, SALL4 3′UTR	175−181	[41]
		miR-188	In Silico, In Vitro	Binds to SALL4 promoter		[56]
	Glioma	miR-15b	In Silico, In Vitro	GCAGCA, SALL4 3′ UTR	175−180	[39]
		miR-16	In Silico, In Vitro	UAGCAGCA, SALL4 3′UTR	175−181	[26]
		miR-103	In Silico, In Vitro	GCAGGA, SALL4 3′ UTR	175−180	[39]
		miR-107	In Silico, In Vitro, In Vivo	GCAGCAU-G-A-CAGG, SALL4 3′UTR	167−180	[17]
		miR-181b	In Silico, In Vitro	ACAUUCA, SALL4 3′UTR	1843−1849	[55]
		miR-195	In Silico, In Vitro	GCAGCA, SALL4 3′ UTR	175−180	[39]
		miR-219	In Silico, In Vitro	UGAUUGU, SALL4 3′ UTR	156−162	[38]
	Liver (HCC)	miR-33b	In Silico, In Vitro	UGCAUUGCU, SALL4 3′UTR	23−29	[43]
		miR-98	In Silico, In Vitro, In Vivo	UGAGGUAG, SALL4 3′UTR	1335−1342	[15]
		miR-296	In Silico, In Vitro, In Vivo	Indirect repression via Brg1		[54]
	Lung	miR-98	In Silico, In Vitro, In Vivo	GAGGUAG, SALL4 3′ UTR	1335−1341	[16]
		miR-205	In Silico, In Vitro, In Vivo	CCUUCAU, SALL4 3′UTR	1280−1286	[60]
		miR-3619	In Silico, In Vitro, In Vivo	UCAGCAGG, SALL4 3′ UTR (HOXA11-AS works as a sponge to compete with SALL4 in binding to miR-3619)	1880−1887	[53]
	Ovarian	miR-98	In Vitro	Full text not available		[14]
microRNAs regulated by SALL4	Liver (HCC)	miR-146a	In Silico, In Vitro, In Vivo	SALL4 binds to miR-146a-5p promoter		[52]
		miR-200c	In Silico, In Vitro, In Vivo	SALL4 binds to miR-200c promoter		[49]

## Data Availability

Not applicable.

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
