# Peer review of "SALL4 and microRNA: The Role of Let-7"

_genes, 2021, doi:10.3390/genes12091301_

Round 1

Reviewer 1 Report

In the REVIEW “SALL4 and microRNA: the role of Let-7
“Liu et al., give a overview current knowledge of SALL4 and miRNAs functions in development and cancer.  The paper is well written. However, the following suggestions are recommended to improve the quality of the manuscript: 

  1. In Materials and Methods authors should provide flow chart including number of records identified, included and excluded, and the reasons for exclusions.
  2. Please add more info about members of let-7 family (including info why mir-98 belonge to let-7 family)
  3. line 174 "A recent study found that Mir-98 levels are.." mir-98
  4. line 290 wrong font used for paragraph 3.6

Author Response

Reviewer 1

“Liu et al., give a overview current knowledge of SALL4 and miRNAs functions in development and cancer.  The paper is well written. However, the following suggestions are recommended to improve the quality of the manuscript:

1).In Materials and Methods authors should provide flowchart including number of records identified, included and excluded, and the reasons for exclusions.

Response: We added a flow chart in the main text.

2).Please add more info about members of let-7 family (including info why mir-98 belong to let-7 family)

Response: we appreciate the reviewer’s comment. The following content has been added in section 3.1: “The human genome encodes a total of 13 let-7 family members with 9 mature miRNAs. Although Let-7a1, a2, a3 are encoded by different transcripts, they produce identical mature sequence; let-7f1 and f2 transcripts also give rise to identical mature miRNA (Chirshev et al. 2019). Mir-98 is considered a member of the let-7 family based on its seed sequence.” and “LIN28 recognition by Let-7 microRNA is mediated by targeting of both the zinc knuckle domain (ZKD), which recognizes a GGAG-like element in the microRNA precursor, as well as the cold shock domain (CSD). The mechanisms of CSD domain binding on the microRNA have recently been characterized, and the (U)GAU motif was shown to be the target sequence (Ustianenko et al. 2018). This motif separates the let-7 microRNAs into two subclasses, including precursors with both CSD and ZKD binding sites (CSD+ ) and precursors with ZKD but no CSD binding sites (CSD-). In vivo recognition of CSD+ precursors by LIN28 have been shown to be more efficient, which corresponds to stronger suppression of these microRNAs in LIN28-activated cells and cancers. CSD+ Let-7 precursors include pre-let-7b, pre-let-7d, pre-letf-1, pre-let-7g, and mir-98. The binding between LIN28 and mir-98 CSD domain has been validated in vivo: loss-of-function mutant of pri-let-7 hairpins from miR-98 where the (U)GAU motif was mutated shows greatly reduced the amount of LIN28 protein binding (Ustianenko et al. 2018).”

3) line 174 "A recent study found that Mir-98 levels are.." mir-98

Response: We have fixed this error.

4) line 290 wrong font used for paragraph 3.6

Response: We have fixed this error.

Reviewer 2 Report

Liu et all wrote this review highlighting the latest evidence regarding the involvement of the transcription factor SALL4 in the regulation and control of the development and progression of numerous types of cancer. The mediation of these effects through Let-7 / Lin-28, and above all, their effectors, miRNAs represent a very important and current approach in the therapy of various cancerous pathologies in humans. In particular, the authors focused their attention by collecting the latest news on HCC and NSCLC which are among the cancerous diseases with the greatest negative outcome. The topic is very interesting and topical, the data have been collected very well and reported with care as well as the references have been appropriately selected, these data are very important for the whole scientific community that is interested in studying the regulatory effects exerted by miRNAs and the possible therapeutic strategies that can derive from them. I have some suggestions to enhance the manuscript even more and make it more complete.

1) Very specific data have been reported regarding the involvement of Let-7 miRNAs and SALL4 in liver cancer, as well as non-small cell lung cancer.

I suggest doing the same thing with other cancers that are just as interesting as gastric, ovarian, colorectal, breast, acute myeloid leukemia and glioma, one paragraph each.

a double check that mir146a acts on NF-kB must be given through the in vitro treatment with a specific antibody that blocks NF-kB or a specific siRNA

2) Numerous miRNAs are involved in these tumor diseases and exert their regulatory action. Some only in specific pathologies others instead in more pathologies. Only a few have been mentioned. It would be very important to report the effects, documented bibliographically, for each one referred to the pathology where its involvement has been reported. Some only in specific pathologies others instead in more pathologies.

3) It would be very important to insert a table showing all the pathologies considered with the miRna involved and their variations next to it, as well as the bibliographic references, so that the reader can immediately have a clear idea for each pathology.

4) Figure 1 should be better detailed, the involvement of miRNAS is noted but it is poor.

5) Finally, a figure with a cartoon should be inserted which represents in detail all the pathways involved in the regulation of SALL4, starting from Let-7 / Lin-28, the miRNAs involved and in particular the possible therapeutic strategies proposed.

Author Response

Reviewer 2

Liu et all wrote this review highlighting the latest evidence regarding the involvement of the transcription factor SALL4 in the regulation and control of the development and progression of numerous types of cancer. The mediation of these effects through Let-7 / Lin-28, and above all, their effectors, miRNAs represent a very important and current approach in the therapy of various cancerous pathologies in humans. In particular, the authors focused their attention by collecting the latest news on HCC and NSCLC which are among the cancerous diseases with the greatest negative outcome. The topic is very interesting and topical, the data have been collected very well and reported with care as well as the references have been appropriately selected, these data are very important for the whole scientific community that is interested in studying the regulatory effects exerted by miRNAs and the possible therapeutic strategies that can derive from them. I have some suggestions to enhance the manuscript even more and make it more complete.

1) Very specific data have been reported regarding the involvement of Let-7 miRNAs and SALL4 in liver cancer, as well as non-small cell lung cancer. I suggest doing the same thing with other cancers that are just as interesting as gastric, ovarian, colorectal, breast, acute myeloid leukemia and glioma, one paragraph each.

Response: We appreciate the reviewer’s feedback. We specifically focused on liver cancer and NSCLC because there are direct experimental evidence showing the regulation of SALL4 by let-7 miRNA in these two cancer types. In other cancers, although there are isolated reports on the roles of either Let-7 or SALL4 in pathophysiology, there are no studies that directly link the functions of the two in other cancer types. We have made this point more clear by adding the following in section 3.4: “In addition to liver cancer and NSCLC, both SALL4 and Let-7 family microRNAs have been shown to play important roles in the pathogenesis of multiple cancer types, such as glioma, ovarian cancer, colorectal cancer, breast cancer, and myeloid neoplasms (Chirshev et al. 2019; Zhang et al. 2015). However, further studies are necessary to explore the direct regulation of SALL4 by Let-7 family members in these cancer types.”

2). a double check that mir146a acts on NF-kB must be given through the in vitro treatment with a specific antibody that blocks NF-kB or a specific siRNA

Response: We appreciate the reviewer’s insight. The authors (Yin et al. 2019) showed that the HCC derived exosome could remodel M2 macrophages and activate NFkB signaling with evidence of increased NFkB protein level by Western blot. In the subsequent experiments, the authors showed mir146 is enriched in the exomes and promote M2 remodeling. However, the authors did not show that mir146a directly leads to increased NFkB expression (either by RNA or by Western blot), which is definitely a shortcoming of the study. Therefore we added the following in the paper to focus on the functional significance of mir146 in promoting M2 remodeling and upstream regulation by SALL4, while pointing out the limited evidence regarding NFkB signalling as suggested by the reviewer in section 3.5: “In addition to miR-200c, SALL4 has also been shown to regulate the expression of miR-146a in HCC (Yin et al. 2019). Treatment with HCC-derived exosomes could remodel M2 macrophages, at least in part by activating the NFkB signaling pathway. miR-146a is highly enriched in the HCC exomes, and its expression is under the direct regulation by SALL4 via binding to the microRNA’s promoter region. Although the authors showed that the expression of miR-146a recapitulates the functional consequences of HCC exome treatment by promoting M2 macrophage remodeling and leading to changes in the expression of IFN-γ, TNF-α, PD-1 and CTLA-4, the precise mechanisms (e.g. the involvement of the NFkB pathway) require further elucidation.”

3) Numerous miRNAs are involved in these tumor diseases and exert their regulatory action. Some only in specific pathologies others instead in more pathologies. Only a few have been mentioned. It would be very important to report the effects, documented bibliographically, for each one referred to the pathology where its involvement has been reported. Some only in specific pathologies others instead in more pathologies.

Response: we appreciate the reviewer’s suggestion. We have added a table to include pertinent information regarding all the pathologies (cancer types), SALL4-related microRNAs, bibliography, seed sequence and summary of experimental evidence as suggested. However, due to space limitation and the scope of this review (focusing only on SALL4-regulatory microRNAs), we did not discuss all the microRNAs that are important for tumor biology.

4) It would be very important to insert a table showing all the pathologies considered with the miRna involved and their variations next to it, as well as the bibliographic references, so that the reader can immediately have a clear idea for each pathology.

Response: Thank you for the suggestions. We have added a table (see above) to expand on the different miRNAs implicated in SALL4 regulation, summary of experimental data, seed sequences, locations of SALL4 binding, as well as bibliography.

5) Figure 1 should be better detailed, the involvement of miRNAS is noted but it is poor.

Response: We have included more detail about SALL4 regulation by microRNAs in figure 1 (now changed to figure 2 because figure 1 will be a flow chart). Please see below for the updated figure 2.

Figure 2. Summary of SALL4-interacting microRNAs: Multiple microRNAs have been shown to regulate SALL4 in a variety of cancer types, including glioma, AML, HCC, gastric cancer, ovarian cancer, lung cancer, colorectal cancer and breast cancer. In addition, SALL4 acts as an upstream regulator of the expression of miR-200c and miR-146a in HCC.

6) Finally, a figure with a cartoon should be inserted which represents in detail all the pathways involved in the regulation of SALL4, starting from Let-7 / Lin-28, the miRNAs involved and in particular the possible therapeutic strategies proposed.

Response: We have added an additional figure (figure 3) to illustrate this. Please see the figure and figure legends below.

Figure 3. Strategies in targeting miR/SALL4 axis in cancer. We can inhibit SALL4 protein expression through miRs by 1) using Lin28 inhibitors (such as Difluoromethylornithine, in clinical trial pipeline) to block its repression on Let 7/miR-98 which in turn decreases SALL4 expression. SALL4 expression causes increased expression of MMP2/9, Fibronectin, N-cadherin, and decreased expression of E-cadherin, which correlate with increased migration/invasion and EMT in an HCC in vivo cancer model. Decreased miR-98 and increased SALL4 expression also leads to increased tumor cell proliferation in vivo and this is presumably due to increased downstream expression of SALL4-regulated oncogenes and decreased tumor suppressor gene expression. This relationship is represented by shaded line because the studies on miR-98/SALL4 axis did not demonstrate a direct epistatic relationship downstream of SALL4; or 2) One study showed that the use of epigenetic drugs such as HDAC inhibitor Entinostat can upregulate miR-205, which in turn suppresses SALL4 expression. This leads to increased cellular viability and decreased apoptosis in vivo consistent with known effect of SALL4 on the expression of oncogenes and tumor suppressor genes, but the study did not further investigate the downstream effectors of SALL4 (relationship represented in shaded line).

Round 2

Reviewer 1 Report

The authors have addressed all concerns raised and the article is good to be published.

Author Response

Reviewer 1 confirmed that we addressed all the comments. 

Reviewer 2 Report

The manuscript has been substantially improved. No more questions